# Inner synchronization of controlled multi-valued logical networks

Yunyun Deng[1,2], Xiaolong Qi[1,2], Yi Liang[1,2,3]*

1 School of Network Security and Information Technology, Yili Normal University, Yining, Xinjiang, People's Republic of China, 2 Key Laboratory of Intelligent Computing Research and Application, Yili Normal University, Yining, Xinjiang, People's Republic of China, 3 State Key Laboratory for Novel Software Technology, Nanjing University, Nanjing, Jiangsu, People's Republic of China

☯ These authors contributed equally to this work.
* 18196993215@163.com

## Abstract

This paper studies the inner synchronization problem of multi-valued logical networks by introducing an external single node, which is also called the driving node. Firstly, for the case where the synchronized state of all network nodes equals the state of the driving node, the inner synchronization model of the controlled multi-valued logical network is established, and a necessary and sufficient condition is obtained using the semi-tensor product. Secondly, based on the above synchronization model, a more general model of inner synchronization is discussed, where all nodes of the multi-valued logical network synchronize with each other, but not necessarily to the same as state of the driving node; inner synchronization criteria are established through theoretical proof. Finally, two simulation examples are given to verify the validity of the conclusions.

## Introduction

In the early 1960s, the distinguished scientists Jacob F and Monod J won the Nobel Prize for Physiology or Medicine for their discovery that each cell contains a set of important regulatory genes. These genes have a remarkable ability to turn other genes on and off with precision, thus playing a crucial regulatory role within the cell[1]. Based on the above discovery, Kauffman further proposed the concept of Boolean networks (BNs) and successfully applied it to the modeling of gene regulatory networks [2]. BNs with their unique logical operation rules accurately describe the interactions between genes and collectively regulate cellular life activities. In BNs, the state of a node is represented by a Boolean variable (1 or 0), and each node updates its state based on the state of its neighbors and logical functions [3,4]. In some practical applications, two-valued logic cannot fully capture the complexity and diversity of logical networks. Consequently, a multi-valued logical network model was proposed, which is a more general form of BNs (binary logical networks). A multi-valued logical network allows its nodes to take values from a finite set $\mathcal{D}_k = \{i/(k-1) \mid$

**Data availability statement:** All relevant data are within the manuscript and its Supporting information files.

**Funding:** Key Program of Yili Normal University for Comprehensive Strength (No.: 22XKZZ16), Project of State Key Laboratory for Novel Software Technology, Nanjing University (No.: KFKT2025B34), Program of the Autonomous Region Tianshan (Youth Top) Talent (No.: 2024TSYCJC0029), the National Natural Science Foundation of China (No.: 62366054).

**Competing interests:** The authors declare no conflict of interest.

$i = 0, 1, \cdots, k-1\}(k \geq 2)$, also called $k$-valued logical network [4–6]. Research on multi-valued logical networks has been widely applied to networked evolutionary games [7], fault diagnosis for gate networks [8], finite automata [9], fault detection of digital circuits [10], principles and technology of multi-valued logic devices [11,12] and other fields. Therefore, it is of great significance to study the behavior characteristics of multi-valued logical networks.

At the beginning of the 21st century, Daizhan Cheng and his research team first proposed the concept of the semi-tensor product (STP) [4] and developed a complete theoretical system. The STP breaks through the limitation of matrix multiplication on dimensionality. At the same time, the STP retains most of the important properties of traditional matrix multiplication, such as the distributive law and associative law. The STP has provided a powerful tool for studying logical networks. In recent years, a great number of research results about BNs have been obtained using the STP, for example, controllability and observability [13–15], stability and stabilization [16,17], optimal control and output tracking [18,19]. Furthermore, research has extended to the dynamic behavior of complex networks, which is mainly related to the coupling relationship (structure) of the network and the dynamic characteristics of network nodes. Due to their own attributes or environmental factors, the relationships between network nodes often lead to different switching modes in network dynamics models. For example, optimization control problem of switching Boolean networks has been studied in [19], while [20] explores disturbance decoupling.

Synchronization is a natural phenomenon that exists in many fields, such as pendulum synchronous swing, synchronous transmission of signals, laser oscillation synchronization, and so on. In the past few decades, research on the synchronization of networks has drawn much attention from scholars because it has many potential applications [21–23]. From the perspective of the dynamic characteristics of network nodes, fractional order calculus has the characteristics of long memory and more degrees of freedom. Complex network control with fractional order is very important and there are also some research works. While allowing partial loss of information, Chen et al. [22] discussed the synchronization of complex networks with fractional order using the properties of generalized fractional calculus and the generalized Laplace transform. Recently, the synchronization of logical networks has become a hot topic, and many important achievements have been obtained [24–29]. In fact, there are different types of synchronization of logical networks, including outer synchronization, inner synchronization, and anti-synchronization [24]. Outer synchronization refers to the synchronization of the corresponding node states between two logical networks, while inner synchronization means that all nodes in a single logical network are synchronized with each other. For instance, the synchronization of two BNs was studied in the drive-response configuration, and a necessary and sufficient criterion was presented using the STP [25]. With asynchronous update rules, the outer synchronization of BNs was investigated, and conditions for the synchronization of asynchronous BNs were established [26]. Liu et al. [27] studied the set stability of generalized asynchronous probabilistic Boolean networks with impulsive effects, they designed an algorithm for the largest invariant set, derived stability criteria, verified global stability/ synchronization via different sets, and illustrated results with

examples. Meng et al. [28] studied the synchronization problem of interconnected high-order multi-valued logical networks. Recently, the concept of approximate synchronization was proposed, and several necessary and sufficient conditions were obtained for global and local approximate synchronization of two coupled multi-valued logical networks [29].

In the above research work on the synchronization, the main focus is on the synchronization of two logical networks (i.e., the outer synchronization). Compared to outer synchronization, research work on inner synchronization is less. Zhang et al. [24] first proposed the concept of inner synchronization of BNs and derived a criterion to achieve it. In [30], the inner synchronization problem of BNs with time delays was investigated, and a necessary and sufficient condition for inner synchronization was derived using the STP. Existing research on the inner synchronization problem of logical networks has been done without external control. However, many logical networks cannot achieve the inner synchronization without external node control. In fact, some systems require external intervention to achieve expected goals. In this paper, we establish two new models for the inner synchronization of multi-valued logical networks with a driving node and investigate two kinds of inner synchronization of controlled multi-valued logical networks. In terms of control methods, existing control methods mainly involve one logical network controlling another logical network; however, we use a single logical node to control a logical network. The first network model achieves inner synchronization where all nodes have the same state as the driving node. In the second network model, the inner synchronization state can be different from the state of the driving node. Necessary and sufficient conditions are derived for both kinds of inner synchronization respectively. Finally, two simulation examples are used to verify the validity of the conclusions.

## 1  Preliminaries

In this thesis, the inner synchronization of multi-valued logical networks is studied using STP. Some commonly used symbols are given below.

$R$ is all real number set; $R^n$ is $n$-dimensional column vector space; $M_{m \times n}$ is a set of $m \times n$ real matrix; $M_n$ is the set of $n$-order square matrix; $I_n$ is a $n$-order identity matrix; $\delta_n^i$ is the $i$-th column of the $n$-order identity matrix; $\otimes$ is the Kronecker product (tensor product); $\ltimes$ is left STP, called the STP for short; $\Delta_n = \{\delta_n^i \mid i = 1, 2, \cdots, n\}$; $\mathrm{Col}(A)$ is the column set of matrix $A$; $\mathrm{Col}(A_i)$ is the $i$-column of matrix $A$. If the column set $\mathrm{Col}(A)$ of a matrix $A$ can be written in the form $\delta_n^i$, then the matrix $A$ is called a logical matrix, in other words, $\mathrm{Col}(A) \subseteq \Delta_n$, denoted by $A \in \mathcal{L}_{n \times m}$.

The definitions, properties and related theorems used in the paper are given below, and the basic knowledge and conclusions are mainly from reference [4].

**Definition 1.1. [4]** (matrix tensor product ) $A = (a_{ij}) \in M_{m \times n}$, $B$ is any matrix, then the Kronecker product of $A$ and $B$ is

$$A \otimes B = \begin{bmatrix} a_{11}B & a_{12}B \cdots a_{1n}B \\ a_{21}B & a_{22}B \cdots a_{2n}B \\ \cdots \\ a_{m1}B & a_{m2}B \cdots a_{mn}B \end{bmatrix}. \tag{1}$$

**Definition 1.2. [4]** (STP) Setting $M \in \boldsymbol{M}_{m \times n}, N \in \boldsymbol{M}_{p \times q}$, the matrix semi-tensor product is

$$C = M \ltimes N = \left( M \otimes I_{l/n} \right) \left( N \otimes I_{l/p} \right), \tag{2}$$

where $l = lcm[n, p]$ is the least common multiple of $n$ and $p$. Obviously, when $n = p$, the product of matrix $M$ and $N$ is transformed into the general matrix product.

Specifically, when $M = \delta_m^i$, $N = \delta_p^j$, $M \ltimes N = \delta_m^i \ltimes \delta_p^j = \delta_{mp}^{(i-1)+j}$.

In the following, we give some propositions on the matrix STP.

**Proposition 1.1. [4]** The following associative law hold when the matrix has appropriate dimensions,

$$F \ltimes (G \ltimes H) = (F \ltimes G) \ltimes H.$$

**Proposition 1.2. [4]** Let $A \in M_{m \times n}$, when $Z \in R^t$ is a column of vectors,

$$(Z \ltimes A) = (I_t \otimes A) \ltimes Z.$$

**Definition 1.3. [4]** A swap matrix $W_{[m,n]}$, which is an $mn \times mn$ matrix, defined as:

$W_{[m,n]} = \delta_{mn}[1, m+1, 2m+1, \ldots, (n-1)m+1, 2, m+2, 2m+2, \ldots, (n-1)m+2, \ldots, m, m+m, 2m+m, \ldots, (n-1)m+m], W_{[m,n]} \in \mathcal{L}_{mn \times mn}$.

**Proposition 1.3. [4]** Let $X = (x_1, x_2, \cdots, x_m)^T$ and $Y = (y_1, y_2, \cdots, y_n)^T$ are column vectors, then $Y \ltimes X = W_{[m,n]} X \ltimes Y, X \ltimes Y = W_{[n,m]} Y \ltimes X$.

**Definition 1.4. [4]** If $\sigma \in \Delta_k$, then $\sigma \ltimes \sigma = M_{r,k}\sigma$, which $M_{r,k} = \begin{bmatrix} \delta_k^1 & 0_k & \cdots & 0_k \\ 0_k & \delta_k^2 & \cdots & 0_k \\ \vdots & & & \vdots \\ 0_k & 0_k & \cdots & \delta_k^k \end{bmatrix}$ is a reduced power matrix,

$M_{r,k} \in \mathcal{L}_{k^2 \times k}$.

Multi-valued logic has a similar structure to Boolean logic. The symbols of multivalued logic are given below.

(1) $\mathcal{D}_k = \{1, (k-2)/(k-1), \cdots, 1/(k-1), 0\}$ is the domain of a $k$-valued logic, where $T = 1 \sim \delta_k^1$, $(k-2)/(k-1) \sim \delta_k^2, \cdots$, $F = 0 \sim \delta_k^k$.

(2) Structure matrix for negation: $\neg$, denoted by $M_\neg = M_{n,k} = \delta_k[k \quad k-1 \cdots 1]$. For example, when $k = 3$, $M_{n,3} = \delta_3[3 \quad 2 \quad 1]$.

Similarly, for conjunction ($\wedge$), disjunction ($\vee$), conditional ($\rightarrow$), we define their corresponding structure matrices, denoted by $M_c$, $M_d$, and $M_i$, respectively, as follows:

$$M_\wedge = M_{c,k} = \delta_k[\underbrace{123 \cdots k}_{k} \underbrace{223 \cdots k}_{k} \underbrace{333 \cdots k}_{k} \cdots \underbrace{kk \cdots k}_{k}],$$

$$M_\vee = M_{d,k} = \delta_k[\underbrace{111 \cdots 1}_{k} \underbrace{222 \cdots 2}_{k} \underbrace{123 \cdots 3}_{k} \cdots \underbrace{123 \cdots k}_{k}],$$

$$M_\rightarrow = M_{i,k} = \delta_k[\underbrace{123 \cdots k}_{k} \underbrace{12 \cdots k-1 \; k-1}_{k} \cdots \underbrace{11 \cdots 1}_{k}].$$

(3) Let $f(x_1, x_2, \cdots, x_n)$ be a $k$-valued logical function. Then, there exists a unique structure matrix $M_f \in \mathcal{L}_{k \times k^n}$ such that $x_i \in \mathcal{D}_k, M_f \in \mathcal{L}_{k \times k^n}$ is the structure matrix of $f(\cdot)$.

There are often the lack of some logical variables by converting a specific logical equation into a general algebraic equation and it is necessary to introduce a special matrix (dumb matrix). In algebraic equations, the sign "$\ltimes$" of the STP is often omitted without ambiguity.

**Proposition 1.4. [4]** A dummy operator $\sigma_d$, defined by $\sigma_d(p, q) = q, \forall p, q \in \mathcal{D}_k$, the structure matrix of the base-$k$ dummy operator $\sigma_d$ is $E_{d,k} = \delta_k \underbrace{I_k I_k \cdots I_k}_{k}$.

It follows from the definition that for any logical variables $X, Y$, then $E_d X Y = Y$ or $E_d W_{[k,k]} X Y = X$.

## 2 Results

The existing research primarily focuses on using one logical network to drive another, aiming for the corresponding nodes between the two networks to achieve synchronization. In contrast, our work achieves inner synchronization within a single logical network by driving it with just one node.

In the section, we establish models of the controlled multi-valued logical networks and study their inner synchronization problems separately. Here is an example of a network that cannot achieve synchronization solely through mutual coupling between nodes.

Consider an example of a three-valued logical network that can not achieve inner synchronization without driving node:

$$\begin{cases} x_1(t+1) = x_1(t) \to x_3(t) \\ x_2(t+1) = x_3(t) \\ x_3(t+1) = x_2(t) \end{cases}. \tag{3}$$

For initial value $(x_1(0), x_2(0), x_3(0)) = (0, 0.5, 0)$, we give the evolution process of the states of each node in the network (3):

$$\{0, 0.5, 0\} \to \{1, 0, 0.5\} \to \{0.5, 0.5, 0\} \to \{0.5, 0, 0.5\} \to \{0.5, 0.5, 0\} \to \cdots.$$

Obviously, without a driving node, the node states in the (3) cannot be synchronized.

In the following, the inner synchronization problem of multi-valued logical networks is analyzed under the control of a single driving node.

### 2.1 Inner synchronization with the same as the driving node

Consider the following dynamic equation of a controlled multi-valued logical network:

$$\begin{cases} x_0(t+1) = f_0 \left( x_0(t) \right) \\ x_1(t+1) = f_1 \left( x_0(t), x_1(t), x_2(t), \cdots, x_n(t) \right) \\ x_2(t+1) = f_2 \left( x_0(t), x_1(t), x_2(t), \cdots, x_n(t) \right) \\ \cdots \\ x_n(t+1) = f_n \left( x_0(t), x_1(t), x_2(t), \cdots, x_n(t) \right) \end{cases}, \tag{4}$$

where $x_i(\cdot) \in \mathcal{D}_k$ is a logical variable for node $i$, $x_0(\cdot)$ is a driving node, $f_i(\cdot) : \mathcal{D}_{k^{n+1}}^{n+1} \to \mathcal{D}_{k^{n+1}}$ is a multi-valued logical function. Let $x(t) = \ltimes_{i=0}^{n} x_i(t)$, for each $f_i(\cdot)$, it corresponds to structure matrix $M_i$ such that it satisfies $x_i(t+1) = M_i x(t)$, $i = 0, 1, 2, \cdots, n$.

Based on Definitions 1.1 and Proposition 1.1, 1.3, multiplying the above $n+1$ equations yields an equivalent algebraic expression for the system as:

$$x(t+1) = Fx(t), \tag{5}$$

where $F \in \mathcal{L}_{k^{n+1} \times k^{n+1}}$ is called a state transfer matrix of the multi-valued logical network, and $\text{Col}_i(F) = \ltimes_{j=1}^{n} \text{Col}_i(M_j)$, $i = 1, 2, \cdots, k^{n+1}$.

Now, give the definition of synchronization of multi-valued logical networks.

**Definition 2.1.** If for all nodes $x_i(\cdot) \in \mathcal{D}_k, i = 1, 2, \cdots, n$ in the multi-valued logical network (4), there exists a positive integer $a$ such that $t \geq a$ and $x_0(t) = x_1(t) = x_2(t) = \cdots = x_n(t)$, the multi-valued logical network (4) is said to achieve inner synchronization with the same as the driving node.

**Theorem 2.1.** The multi-valued logical network (4) achieves inner synchronization with the same as the driving node if and only if there exists a positive integer $a$ such that $\text{Col}\,(F^a) \subseteq \Gamma_1$, where

$$\Gamma_1 = \left\{ \delta_{k^{n+1}}^1, \delta_{k^{n+1}}^{\frac{k^{n+1}-1}{k-1}+1}, \delta_{k^{n+1}}^{\frac{2(k^{n+1}-1)}{k-1}+1}, \cdots, \delta_{k^{n+1}}^{\frac{(m-1)(k^{n+1}-1)}{k-1}+1}, \cdots, \delta_{k^{n+1}}^{k^{n+1}} \right\},$$

$$m = 1, 2, \cdots, k,$$

(6)

and $F$ is defined by (5).

Proof. (Sufficiency) Suppose that there exists a positive integer $a$ such that $\text{Col}\,(F^a) \subseteq \Gamma_1$. Apparently $\text{Col}\,(F^{a+l}) \subseteq \cdots \subseteq \text{Col}\,(F^{a+1}) \subseteq \text{Col}\,(F^a) \subseteq \Gamma_1$, $l \geq 1$ is an integer. When $t \geq a$, it is easy to calculate that $x(t) = F^t x_0(0) \ltimes x(0) \in \text{Col}\,(F^a)$. Therefore, there exists $1 \leq m_0 \leq k$ such that $x(t) = \delta_{k^{n+1}}^{\frac{(m_0-1)(k^{n+1}-1)}{k-1}+1}$ $(t \geq a)$. Also because by Definition 1.2, we can obtain $\left(\delta_k^{m_0}\right)^{n+1} = \delta_{k^{n+1}}^{\frac{(m_0-1)(k^{n+1}-1)}{k-1}+1}$, for any initial value, $x(t) = \delta_{k^{n+1}}^{\frac{(m_0-1)(k^{n+1}-1)}{k-1}+1} = \left(\delta_k^{m_0}\right)^{n+1}$, so when $t \geq a$, $x_0(t) = x_1(t) = x_2(t) = \cdots = x_n(t) = \delta_k^{m_0}$. That is to say, the controlled multi-valued logical network achieves inner synchronization with the same as the driving node.

(Necessity) Assuming that at moment $a$, the controlled multi-valued logical network achieves inner synchronization with the same as the driving node, then there exists $1 \leq m \leq k$ such that $x_0(a) = x_1(a) = x_2(a) \cdots = x_n(a) = \delta_k^m$, and $\left(\delta_k^m\right)^{n+1} = \delta_{k^{n+1}}^{\frac{(m-1)(k^{n+1}-1)}{k-1}+1}$, from $x(a) = F^a \ltimes_{i=0}^n x_i(0) = \delta_{k^{n+1}}^{\frac{(m-1)(k^{n+1}-1)}{k-1}+1}$. Since the initial values of the multi-valued logical network are randomly selected, we have $\text{Col}\,(F^a) = \delta_{k^{n+1}}^{\frac{(m-1)(k^{n+1}-1)}{k-1}+1}$. Hence, $\text{Col}\,(F^a) \subseteq \Gamma_1$, $m = 1, 2, \cdots, k$. Theorem 2.1 is proved.

This article studies deterministic logical networks. If Eq (4) is subjected to external interference, it can be divided into two cases: when only the driving node is subjected to external disturbances, the conclusion of Theorem 2.1 still holds. Because the evolution process of network nodes is only related to the initial value of the driving node, and not to their subsequent states. If the driven network is disturbed, Theorem 2.1 will not hold, and this issue also needs further exploration.

## 2.2 General inner synchronization

In some cases, all nodes in the controlled multi-valued logical network synchronize with each other, but may not be in the same state as the driving node.

The controlled multi-valued logical network is as follows:

$$\begin{cases} x_0(t+1) = f_0\left(x_0(t)\right) \\ x_1(t+1) = f_1\left(x_0(t), x_1(t), x_2(t), \cdots, x_n(t)\right) \\ x_2(t+1) = f_2\left(x_0(t), x_1(t), x_2(t), \cdots, x_n(t)\right) \\ \cdots \\ x_n(t+1) = f_n\left(x_0(t), x_1(t), x_2(t), \cdots, x_n(t)\right) \end{cases}, \quad (7)$$

where $x_i(\cdot) \in \mathcal{D}_k$ is the state logical variable of node $i$, $x_0(\cdot)$ is a driving node, $f_i(\cdot) : \mathcal{D}_k^{n+1} \to \mathcal{D}_k (i = 0, 1, 2, \cdots, n)$ is the multi-valued logical function. There exists a corresponding matrix $M_i$ for each $f_i(\cdot)$, such that it satisfies $x_i(t+1) = M_i x(t)$. Same as the above analysis, we obtain an equivalent algebraic expression for the system as:

$$x(t+1) = Fx(t), \quad (8)$$

where $F \in \mathcal{L}_{k^{n+1} \times k^{n+1}}$, and $\text{Col}_i(F) = \ltimes_{j=1}^n \text{Col}_i\left(M_j\right), i = 1, 2, \cdots, k^{n+1}$.

**Definition 2.2.** If for all nodes $x_i(\cdot) \in \mathcal{D}_k, i = 1, 2, \cdots, n$ in the controlled multi-valued logical network (7), there exists a positive integer $a$ such that when $t \geq a$ and $x_1(t) = x_2(t) = \cdots = x_n(t)$, the controlled multi-valued logical network (7) is said to achieve inner synchronization.

**Theorem 2.2.** Multi-valued logical network (7) achieves inner synchronization if and only if there exists a positive integer $a$ such that $\mathrm{Col}(F^a) \subseteq \Gamma_2$, where

$$
\begin{aligned}
\Gamma_2 = \Big\{ & \delta_{k^{n+1}}^1, \delta_{k^{n+1}}^{1+\frac{k^n-1}{k-1}}, \delta_{k^{n+1}}^{1+2\times\frac{k^n-1}{k-1}}, \dots, \delta_{k^{n+1}}^{1+(i-1)\frac{k^n-1}{k-1}}, \dots, \\
& \delta_{k^{n+1}}^{k^n}, \delta_{k^{n+1}}^{k^n+1}, \delta_{k^{n+1}}^{k^n+1+\frac{k^n-1}{k-1}}, \delta_{k^{n+1}}^{k^n+1+2\times\frac{k^n-1}{k-1}}, \dots, \delta_{k^{n+1}}^{k^n+1+(i-1)\frac{k^n-1}{k-1}}, \dots, \\
& \delta_{k^{n+1}}^{2\times k^n}, \delta_{k^{n+1}}^{2\times k^n+1}, \delta_{k^{n+1}}^{2\times k^n+1+\frac{k^n-1}{k-1}}, \delta_{k^{n+1}}^{2\times k^n+1+2\times\frac{k^n-1}{k-1}}, \dots, \delta_{k^{n+1}}^{2\times k^n+1+(i-1)\frac{k^n-1}{k-1}}, \\
& \dots, \delta_{k^{n+1}}^{3\times k^n}, \dots, \delta_{k^{n+1}}^{(j-1)\times k^n+1}, \delta_{k^{n+1}}^{(j-1)\times k^n+1+\frac{k^n-1}{k-1}}, \delta_{k^{n+1}}^{(j-1)\times k^n+1+2\times\frac{k^n-1}{k-1}}, \dots, \\
& \delta_{k^{n+1}}^{(j-1)\times k^n+1+(i-1)\frac{k^n-1}{k-1}}, \dots, \delta_{k^{n+1}}^{j\times k^n}, \dots, \delta_{k^{n+1}}^{(k-1)\times k^n+1}, \delta_{k^{n+1}}^{(k-1)\times k^n+1+\frac{k^n-1}{k-1}}, \\
& \delta_{k^{n+1}}^{(k-1)\times k^n+1+2\times\frac{k^n-1}{k-1}}, \dots, \delta_{k^{n+1}}^{(k-1)\times k^n+1+(i-1)\frac{k^n-1}{k-1}}, \dots, \delta_{k^{n+1}}^{k^{n+1}} \Big\}
\end{aligned}
\tag{9}
$$

and $F$ is defined by Eq (8). On the right side of $\Gamma_2$, there are a total of $k^2$ terms, $k$ groups, each consisting of $k$ terms.

Proof. Let $\bar{x}(t) = x_0(t) \ltimes x(t), x(t) = \ltimes_{i=1}^n x_i(t)$, then $\bar{x}(t) = F^t x_0(0) \ltimes x(0) \in \mathrm{Col}(F^a), t \geq a$.

(Sufficiency) Suppose there exists a positive integer $a$ such that $\mathrm{Col}(F^a) \subseteq \Gamma_2$. For $t \geq a$ and $l \geq 1$, we have $\mathrm{Col}(F^{a+l}) \subseteq \cdots \subseteq \mathrm{Col}(F^{a+1}) \subseteq \mathrm{Col}(F^a) \subseteq \Gamma_2$.

Case 1. $x_0(t) = \delta_k^1$. Based on Definition 1.2,

$$\bar{x}(t) \notin \{\delta_{k^{n+1}}^{k^n+1}, \delta_{k^{n+1}}^{k^n+1+\frac{k^n-1}{k-1}}, \delta_{k^{n+1}}^{k^n+1+2\times\frac{k^n-1}{k-1}}, \dots, \delta_{k^{n+1}}^{k^n+1+(i-1)\frac{k^n-1}{k-1}}, \dots, \delta_{k^{n+1}}^{2\times k^n}, \delta_{k^{n+1}}^{2\times k^n+1}, \delta_{k^{n+1}}^{2\times k^n+1+\frac{k^n-1}{k-1}}, \dots, \delta_{k^{n+1}}^{2\times k^n+1+2\times\frac{k^n-1}{k-1}}, \dots, \delta_{k^{n+1}}^{2\times k^n+1+(i-1)\frac{k^n-1}{k-1}}, \dots,$$

$$\delta_{k^{n+1}}^{3\times k^n}, \dots, \delta_{k^{n+1}}^{(j-1)\times k^n+1}, \delta_{k^{n+1}}^{(j-1)\times k^n+1+\frac{k^n-1}{k-1}}, \delta_{k^{n+1}}^{(j-1)\times k^n+1+2\times\frac{k^n-1}{k-1}}, \dots, \delta_{k^{n+1}}^{(j-1)\times k^n+1+(i-1)\frac{k^n-1}{k-1}}, \dots, \delta_{k^{n+1}}^{j\times k^n}, \dots, \delta_{k^{n+1}}^{(k-1)\times k^n+1}, \delta_{k^{n+1}}^{(k-1)\times k^n+1+\frac{k^n-1}{k-1}},$$

$$\delta_{k^{n+1}}^{(k-1)\times k^n+1+2\times\frac{k^n-1}{k-1}}, \dots, \delta_{k^{n+1}}^{(k-1)\times k^n+1+(i-1)\frac{k^n-1}{k-1}}, \dots, \delta_{k^{n+1}}^{k^{n+1}}\}.$$

Otherwise, let $x(t) = \delta_{k^n}^m$, $1 \leq m \leq k^n$, then $\bar{x}(t) = \delta_{k^{n+1}}^m$. It is easy to get that $\bar{x}(t) \notin \{\delta_{k^{k^n+1}}^{k^n}, \delta_{k^{n+1}}^{k^n+1+\frac{k^n-1}{k-1}}, \delta_{k^{n+1}}^{k^n+1+2\times\frac{k^n-1}{k-1}}, \dots,$

$$\delta_{k^{n+1}}^{k^n+1+(i-1)\frac{k^n-1}{k-1}}, \dots, \delta_{k^{n+1}}^{2\times k^n}, \delta_{k^{n+1}}^{2\times k^n+1}, \delta_{k^{n+1}}^{2\times k^n+1+\frac{k^n-1}{k-1}}, \delta_{k^{n+1}}^{2\times k^n+1+2\times\frac{k^n-1}{k-1}}, \dots, \delta_{k^{n+1}}^{2\times k^n+1+(i-1)\frac{k^n-1}{k-1}}, \dots, \delta_{k^{n+1}}^{3\times k^n}, \dots, \delta_{k^{n+1}}^{(j-1)\times k^n+1}, \delta_{k^{n+1}}^{(j-1)\times k^n+1+\frac{k^n-1}{k-1}},$$

$$\delta_{k^{n+1}}^{(j-1)\times k^n+1+2\times\frac{k^n-1}{k-1}}, \dots, \delta_{k^{n+1}}^{(j-1)\times k^n+1+(i-1)} K_{k-1}^{k^n-1}, \delta_{k^{n+1}}^{j\times k^n}, \dots, \delta_{k^{n+1}}^{(k-1)\times k^n+1}, \delta_{k^{n+1}}^{(k-1)\times k^n+1+\frac{k^n-1}{k-1}}, \delta_{k^{n+1}}^{(k-1)\times k^n+1+2\times\frac{k^n-1}{k-1}}, \dots, \delta_{k^{n+1}}^{(k-1)\times k^n+1+(i-1)\frac{k^n-1}{k-1}}, \dots,$$

$\delta_{k^{n+1}}^{k^{n+1}}\}$. Hence, $\bar{x}(t) \in \{\delta_{k^{n+1}}^1, \delta_{k^{n+1}}^{1+\frac{k^n-1}{k-1}}, \delta_{k^{n+1}}^{1+2\times\frac{k^n-1}{k-1}}, \dots, \delta_{k^{n+1}}^{1+(i-1)\frac{k^n-1}{k-1}}, \dots, \delta_{k^{n+1}}^{k^n}\}$ When $\bar{x}(t) = \delta_{k^{n+1}}^1$. It is easy to calculate that $x(t) = \delta_{k^n}^1$, that is, $x_i(t) = \delta_k^1, 1 \leq i \leq n$. For the same reason, when $\bar{x}(t) = \delta_{k^{n+1}}^{1+\frac{k^n-1}{k-1}}, \bar{x}(t) = \delta_{k^{n+1}}^{1+(i-1)\frac{k^n-1}{k-1}}$ and $\bar{x}(t) = \delta_{k^{n+1}}^{k^n}$, we can obtain $x(t) = \delta_{k^n}^{1+\frac{n^n-1}{k-1}} (x_i(t) = \delta_k^2), x(t) = \delta_{k^n}^{1+(i-1)\frac{k^n-1}{k-1}} (x_i(t) = \delta_k^i)$ and $x(t) = \delta_{k^n}^{k^n} (x_i(t) = \delta_k^k), 1 \leq i \leq n$, respectively. In other words, for $x_0(t) = \delta_k^1$ and multi-valued logical network (7) can achieve inner synchronization.

Case 2. $x_0(t) = \delta_k^2$.

$$\bar{x}(t) \notin \{\delta_{k^{n+1}}^1, \delta_{k^{n+1}}^{1+\frac{k^n-1}{k-1}}, \delta_{k^{n+1}}^{1+2\times\frac{k^n-1}{k-1}}, \dots, \delta_{k^{n+1}}^{1+(i-1)\frac{k^n-1}{k-1}}, \dots, \delta_{k^{n+1}}^{k^n}, \delta_{k^{n+1}}^{2\times k^n+1}, \delta_{k^{n+1}}^{2\times k^n+1+\frac{k^n-1}{k-1}}, \delta_{k^{n+1}}^{2\times k^n+1+2\times\frac{k^n-1}{k-1}}, \dots, \delta_{k^{n+1}}^{2\times k^n+1+(i-1)\frac{k^n-1}{k-1}}, \dots,$$

$$\delta_{k^{n+1}}^{3\times k^n}, \dots \delta_{k^{n+1}}^{(j-1)\times k^n+1}, \delta_{k^{n+1}}^{(j-1)\times k^n+1+\frac{k^n-1}{k-1}}, \delta_{k^{n+1}}^{(j-1)\times k^n+1+2\times\frac{k^n-1}{k-1}}, \dots, \delta_{k^{n+1}}^{(j-1)\times k^n+1+(i-1)\frac{k^n-1}{k-1}}, \dots, \delta_{k^{n+1}}^{j\times k^n}, \dots, \delta_{k^{n+1}}^{(k-1)\times k^n+1}, \delta_{k^{n+1}}^{(k-1)\times k^n+1+\frac{k^n-1}{k-1}},$$

$$\delta_{k^{n+1}}^{(k-1)\times k^n+1+2\times\frac{k^n-1}{k-1}}, \dots, \delta_{k^{n+1}}^{(k-1)\times k^n+1+(i-1)\frac{k^n-1}{k-1}}, \dots, \delta_{k^{n+1}}^{k^{n+1}}\}.$$ Otherwise, let's assume that $x(t) = \delta_{k^n}^m (1 \leq m \leq k^n)$, then, $\bar{x}(t) = \delta_{k^n}^{k^n+m}$.

Obviously, $\bar{x}(t) \notin \{\delta_{k^{n+1}}^1, \delta_{k^{n+1}}^{1+\frac{k^n-1}{k-1}}, \delta_{k^{n+1}}^{1+2\times\frac{k^n-1}{k-1}}, \dots, \delta_{k^{n+1}}^{1+(i-1)\frac{k^n-1}{k-1}}, \dots, \delta_{k^{n+1}}^{k^n}, \delta_{k^{n+1}}^{2\times k^n+1}, \delta_{k^{n+1}}^{2\times k^n+1+\frac{k^n-1}{k-1}}, \delta_{k^{n+1}}^{2\times k^n+1+2\times\frac{k^n-1}{k-1}}, \dots, \delta_{k^{n+1}}^{2\times k^n+1+(i-1)\frac{k^n-1}{k-1}}, \dots,$

$\delta_{k^n+1}^{3\times k^n}, \dots \delta_{k^n+1}^{(j-1)\times k^n+1}, \delta_{k^n+1}^{(j-1)\times k^n+1+\frac{k^n-1}{k-1}}, \delta_{k^n+1}^{(j-1)\times k^n+1+2\times\frac{k^n-1}{k-1}}, \dots, \delta_{k^n+1}^{(j-1)\times k^n+1+(i-1)\frac{k^n-1}{k-1}}, \dots, \delta_{k^n+1}^{j\times k^n}, \dots, \delta_{k^n+1}^{(k-1)\times k^n+1}, \delta_{k^n+1}^{(k-1)\times k^n+1+\frac{k^n-1}{k-1}},$
$\delta_{k^n+1}^{(k-1)\times k^n+1+2\times\frac{k^n-1}{k-1}}, \dots, \delta_{k^n+1}^{(k-1)\times k^n+1+(i-1)\frac{k^n-1}{k-1}}, \dots, \delta_{k^n+1}^{k^{n+1}}\}$. Hence, $\bar{x}(t) \in \{\delta_{k^n+1}^{k^n+1}, \delta_{k^n+1}^{k^n+1+\frac{k^n-1}{k-1}}, \delta_{k^n+1}^{k^n+1+2\times\frac{k^n-1}{k-1}}, \delta_{k^n+1}^{k^n+1+(i-1)\frac{k^n-1}{k-1}}, \dots,$
$\delta_{k^n+1}^{2\times k^n}\}$. Similarly, when $\bar{x}(t) = \delta_{k^n+1}^{k^n+1}, \bar{x}(t) = \delta_{k^n+1}^{k^n+1+\frac{k^n-1}{k-1}}, \bar{x}(t) = \delta_{k^n+1}^{k^n+1+(i-1)\frac{k^n-1}{k-1}}$ and $\bar{x}(t) = \delta_{k^n+1}^{2\times k^n}$, it easy to get $x(t) = \delta_{k^n}^1 (x_i(t) = \delta_k^1)$,
$x(t) = \delta_{k^n}^{1+\frac{k^n-1}{k-1}} (x_i(t) = \delta_k^2)$, $x(t) = \delta_{k^n}^{1+(i-1)\frac{k^n-1}{k-1}} (x_i(t) = \delta_k^i)$ and $x(t) = \delta_{k^n}^{k^n} (x_i(t) = \delta_k^k)$, $1 \le i \le n$.

Case $j$. $x_0(t) = \delta_k^j, 3 \le j \le k-1$.

$\bar{x}(t) \notin \{\delta_{k^n+1}^1, \delta_{k^n+1}^{1+\frac{k^n-1}{k-1}}, \delta_{k^n+1}^{1+2\times\frac{k^n-1}{k-1}}, \dots, \delta_{k^n+1}^{1+(i-1)\frac{k^n-1}{k-1}}, \dots, \delta_{k^n+1}^{k^n}, \delta_{k^n+1}^{k^n+1}, \delta_{k^n+1}^{k^n+1+\frac{k^n-1}{k-1}}, \delta_{k^n+1}^{k^n+1+2\times\frac{k^n-1}{k-1}}, \dots, \delta_{k^n+1}^{k^n+1+(i-1)\frac{k^n-1}{k-1}}, \dots, \delta_{k^n+1}^{2\times k^n}, \delta_{k^n+1}^{2\times k^n+1},$
$\delta_{k^n+1}^{2\times k^n+1+\frac{k^n-1}{k-1}}, \delta_{k^n+1}^{2\times k^n+1+2\times\frac{k^n-1}{k-1}}, \dots, \delta_{k^n+1}^{2\times k^n+1+(i-1)\frac{k^n-1}{k-1}}, \dots, \delta_{k^n+1}^{3\times k^n}, \dots, \delta_{k^n+1}^{(k-1)\times k^n+1}, \delta_{k^n+1}^{(k-1)\times k^n+1+\frac{k^n-1}{k-1}}, \delta_{k^n+1}^{(k-1)\times k^n+1+2\times\frac{k^n-1}{k-1}}, \dots,$
$\delta_{k^n+1}^{(k-1)\times k^n+1+(i-1)\frac{k^n-1}{k-1}}, \dots, \delta_{k^n+1}^{k^{n+1}}\}$. Otherwise, let's assume that $x(t) = \delta_{k^n}^m (1 \le m \le k^n)$, then, $\bar{x}(t) = \delta_{k^n+1}^{k^n+m}$. Obviously, $\bar{x}(t) \notin$
$\{\delta_{k^n+1}^1, \delta_{k^n+1}^{1+\frac{k^n-1}{k-1}}, \delta_{k^n+1}^{1+2\frac{k^n-1}{k-1}}, \dots, \delta_{k^n+1}^{1+(i-1)\frac{k^n-1}{k-1}}, \dots, \delta_{k^n+1}^{k^n}, \delta_{k^n+1}^{k^n+1}, \delta_{k^n+1}^{k^n+1+\frac{k^n-1}{k-1}}, \delta_{k^n+1}^{k^n+1+2\times\frac{k^n-1}{k-1}}, \dots, \delta_{k^n+1}^{k^n+1+(i-1)\frac{k^n-1}{k-1}}, \dots, \delta_{k^n+1}^{2\times k^n+1}, \delta_{k^n+1}^{2\times k^n+1+\frac{k^n-1}{k-1}},$
$\delta_{k^n+1}^{2\times k^n+1+2\times\frac{k^n-1}{k-1}}, \dots, \delta_{k^n+1}^{2\times k^n+1+(i-1)\frac{k^n-1}{k-1}}, \dots, \delta_{k^n+1}^{3\times k^n}, \dots, \delta_{k^n+1}^{(k-1)\times k^n+1}, \delta_{k^n+1}^{(k-1)\times k^n+1+\frac{k^n-1}{k-1}}, \delta_{k^n+1}^{(k-1)\times k^n+1+2\times\frac{k^n-1}{k-1}}, \dots, \delta_{k^n+1}^{(k-1)\times k^n+1+(i-1)\frac{k^n-1}{k-1}}, \dots,$
$\delta_{k^n+1}^{k^{n+1}}\}$. Hence, $\bar{x}(t) \in \{\delta_{k^n+1}^{(j-1)\times k^n+1}, \delta_{k^n+1}^{(j-1)\times k^n+1+\frac{k^n-1}{k-1}}, \delta_{k^n+1}^{(j-1)\times k^n+1+2\times\frac{k^n-1}{k-1}}, \dots, \delta_{k^n+1}^{(j-1)\times k^n+1+(i-1)\frac{k^n-1}{k-1}}, \dots, \delta_{k^n+1}^{j\times k^n}\}$. Similarly, when
$\bar{x}(t) = \delta_{k^n+1}^{(j-1)\times k^n+1}, \bar{x}(t) = \delta_{k^n+1}^{(j-1)\times k^n+1+\frac{k^n-1}{k-1}}, \bar{x}(t) = \delta_{k^n+1}^{(j-1)\times k^n+1+(i-1)\frac{k^n-1}{k-1}}$ and $\bar{x}(t) = \delta_{k^n+1}^{j\times k^n}$, it easy to get $x(t) = \delta_{k^n}^1 (x_i(t) = \delta_k^1)$,
$x(t) = \delta_{k^n}^{1+\frac{k^n-1}{k-1}} (x_i(t) = \delta_k^2)$, $x(t) = \delta_{k^n}^{1+(i-1)\frac{k^n-1}{k-1}} (x_i(t) = \delta_k^i)$ and $x(t) = \delta_{k^n}^{k^n} (x_i(t) = \delta_k^k)$, $1 \le i \le n$.

Case $k$. $x_0(t) = \delta_k^k$.

Same as the above analysis, it is easy to get that $\bar{x}(t) \notin \{\delta_{k^n+1}^1, \delta_{k^n+1}^{1+\frac{k^n-1}{k-1}}, \delta_{k^n+1}^{1+2\times\frac{k^n-1}{k-1}}, \dots, \delta_{k^n+1}^{1+(i-1)\frac{k^n-1}{k-1}}, \dots, \delta_{k^n+1}^{k^n}, \delta_{k^n+1}^{k^n+1}, \delta_{k^n+1}^{k^n+1+\frac{k^n-1}{k-1}},$
$\delta_{k^n+1}^{k^n+1+2\times\frac{k^n-1}{k-1}}, \dots, \delta_{k^n+1}^{k^n+1+(i-1)\frac{k^n-1}{k-1}}, \dots, \delta_{k^n+1}^{2\times k^n}, \delta_{k^n+1}^{2\times k^n+1}, \delta_{k^n+1}^{2\times k^n+1+\frac{k^n-1}{k-1}}, \delta_{k^n+1}^{2\times k^n+1+2\times\frac{k^n-1}{k-1}}, \dots, \delta_{k^n+1}^{2\times k^n+1+(i-1)\frac{k^n-1}{k-1}}, \dots, \delta_{k^n+1}^{3\times k^n}, \dots, \delta_{k^n+1}^{(j-1)\times k^n+1},$
$\delta_{k^n+1}^{(j-1)\times k^n+1+\frac{k^n-1}{k-1}}, \delta_{k^n+1}^{(j-1)\times k^n+1+2\times\frac{k^n-1}{k-1}}, \dots, \delta_{k^n+1}^{(j-1)\times k^n+1+(i-1)\frac{k^n-1}{k-1}}, \dots, \delta_{k^n+1}^{j\times k^n}, \dots\}$. $\bar{x}(t) \in \{\delta_{k^n+1}^{(k-1)\times k^n+1}, \delta_{k^n+1}^{(k-1)\times k^n+1+\frac{k^n-1}{k-1}}, \delta_{k^n+1}^{(k-1)\times k^n+1+2\frac{k^n-1}{k-1}}, \dots,$
$\delta_{k^n+1}^{(k-1)\times k^n+1+(i-1)\frac{k^n-1}{k-1}}, \dots, \delta_{k^n+1}^{k^{n+1}}\}$. When $\bar{x}(t) = \delta_{k^n+1}^{(k-1)\times k^n+1}, \bar{x}(t) = \delta_{k^n+1}^{(k-1)\times k^n+1+\frac{k^n-1}{k-1}}, \bar{x}(t) = \delta_{k^n+1}^{(k-1)\times k^n+1+(i-1)\frac{k^n-1}{k-1}}$ and $\bar{x}(t) = \delta_{k^n+1}^{k^{n+1}}$, We
have $x(t) = \delta_{k^n}^1 (x_i(t) = \delta_k^1)$, $x(t) = \delta_{k^n}^{1+\frac{k^n-1}{k-1}} (x_i(t) = \delta_k^2)$, $x(t) = \delta_{k^n}^{1+(i-1)\frac{k^n-1}{k-1}} (x_i(t) = \delta_k^i)$ and $x(t) = \delta_{k^n}^{k^n} (x_i(t) = \delta_k^k)$, $1 \le i \le n$.

For the above $k$ cases, the multi-valued logical network (7) can achieve inner synchronization.

(Necessity) Assuming that the multi-valued logical network (7) is already inner synchronized at moment $a$.

Case 1. $x_0(t) = \delta_k^1$. For $t \ge a$, there exist $k$ cases as follows:

When $n$ nodes in the network (7) are synchronized to $\delta_k^1$, that is $x_i(a) = \delta_k^1 (1 \le i \le n)$. It is easy to calculate that
$x(a) = \delta_{k^n}^1$, so $\bar{x}(a) = F^a \ltimes_{i=0}^n x_i(0) = \delta_{k^n+1}^1$. Similarly, When $x_i(a) = \delta_k^2, x_i(a) = \delta_k^i, x_i(a) = \delta_k^k, 1 \le i \le n, \bar{x}(a) = \delta_{k^n+1}^{1+\frac{k^n-1}{k-1}},$
$\bar{x}(a) = \delta_{k^n+1}^{1+(i-1)\frac{h^n-1}{k-1}}, \bar{x}(a) = \delta_{k^n+1}^{k^n}$, respectively.

Case 2. $x_0(t) = \delta_k^2$. When $x_i(a) = \delta_k^1, x_i(a) = \delta_k^2, x_i(a) = \delta_k^i$ and $x_i(a) = \delta_k^k$, $1 \le i \le n$, we can obtain $\bar{x}(a) = \delta_{k^n+1}^{k^n+1}$,
$\bar{x}(a) = \delta_{k^n+1}^{k^n+1+\frac{k^n-1}{k-1}}, \bar{x}(a) = \delta_{k^n+1}^{k^n+1+(i-1)\frac{k^n-1}{k-1}}$ and $\bar{x}(a) = \delta_{k^n+1}^{2\times k^n}$, respectively.

Case $j$. $x_0(t) = \delta_k^j (3 \le j \le k-1)$. When $x_i(a) = \delta_k^1, x_i(a) = \delta_k^2, x_i(a) = \delta_k^i$ and $x_i(a) = \delta_k^k$, $1 \le i \le n$, we can obtain
$\bar{x}(a) = \delta_{k^n+1}^{(j-1)k^n+1}, \bar{x}(a) = \delta_{k^n+1}^{(j-1)k^n+1+\frac{k^n-1}{k-1}}, \bar{x}(a) = \delta_{k^n+1}^{(j-1)\times k^n+1+(i-1)\frac{k^n-1}{k-1}}$ and $\bar{x}(a) = \delta_{k^n+1}^{j\times k^n}$, respectively.

Case $k$. $x_0(t) = \delta_k^k$. Similarly, when $x_i(a) = \delta_k^1, x_i(a) = \delta_k^2, x_i(a) = \delta_k^i$ and $x_i(a) = \delta_k^k, 1 \le i \le n$, we have $\bar{x}(a) = \delta_{k^n+1}^{(k-1)\times k^n+1}$,
$\bar{x}(a) = \delta_{k^n+1}^{(k-1)\times k^n+1+\frac{k^n-1}{k-1}}, \bar{x}(a) = \delta_{k^n+1}^{(k-1)\times k^n+1+(i-1)\frac{k^n-1}{k-1}}$ and $\bar{x}(a) = \delta_{k^n+1}^{k^{n+1}}$.

To sum up, since the initial values of multi-valued logical network (7) are randomly generated, so it can be obtained that $\mathrm{Col}(F^a) \subseteq \Gamma_2$. Theorem 2.2 is proved.

As a special case of the Theorem 2.2, when $k = 2$, or $k = 3$, Eq. (9) correspond to the following forms: $Col(F^a) \subseteq \left\{\delta_{2^{n+1}}^1, \delta_{2^{n+1}}^{2^n}, \delta_{2^{n+1}}^{2^n+1}, \delta_{2^{n+1}}^{2^{n+1}}\right\}$ and $Col(F^a) \subseteq \left\{\delta_{3^{n+1}}^1, \delta_{3^{n+1}}^{\frac{3^n+1}{2}}, \delta_{3^{n+1}}^{3^n}, \delta_{3^{n+1}}^{3^n+1}, \delta_{3^{n+1}}^{\frac{3^{n+1}+1}{2}}, \delta_{3^{n+1}}^{2\times3^n}, \delta_{3^{n+1}}^{2\times3^n+1}, \delta_{3^{n+1}}^{\frac{3^{n+1}+2\times3^n+1}{2}}, \delta_{3^{3^{n+1}}}^{3^{n+1}}\right\}$.

## 3 Simulation examples

In order to verify the effectiveness of the proposed inner synchronization models of multi-valued logical networks, two simulation examples are given, respectively.

**Example 3.1**. Consider the above three-valued logical network (3). From the above analysis, without a driving node, the network cannot achieve inner synchronization.

By adding a driving node $x_0(t)$ to the three-valued logical network (3), the controlled three-valued logical network is:

$$\begin{cases} x_0(t+1) = x_0(t) \\ x_1(t+1) = x_0(t) \wedge (x_1(t) \rightarrow x_3(t)) \\ x_2(t+1) = x_0(t) \vee x_3(t) \\ x_3(t+1) = x_0(t) \wedge x_2(t) \end{cases} \tag{10}$$

Converting the above equation into the following algebraic form using STP ($k = 3$):

$$\begin{cases} x_0(t+1) = x_0(t) \\ x_1(t+1) = M_c x_0(t) M_i x_1(t) x_3(t) \\ x_2(t+1) = M_d x_0(t) x_3(t) \\ x_3(t+1) = M_c x_0(t) x_2(t) \end{cases} \tag{11}$$

Let $x(t) = \ltimes_{i=0}^3 x_i(t)$, then based on Propositions 1.1-1.4 and Definition 1.3, 1.4, the above equation can be transformed into the following form:

$x(t+1) = x_0(t+1)x_1(t+1)x_2(t+1)x_3(t+1)$
$= x_0(t)M_c x_0(t)M_i x_1(t)x_3(t)M_d x_0(t)x_3(t)M_c x_0(t)x_2(t)$
$= (I_3 \otimes M_c)M_r(I_3 \otimes M_i)(I_{27} \otimes M_d)(I_{243} \otimes M_c)x_0(t)x_1(t)x_3(t)x_0(t)\,x_3(t)x_0(t)x_2(t)$
$= (I_3 \otimes M_c)M_r(I_3 \otimes M_i)(I_{27} \otimes M_d)(I_{243} \otimes M_c)W_{[3,27]}M_r(I_9 \otimes M_r)W_{[3,27]}\,M_r(I_9 \otimes W_{[3,3]})x_0(t)x_1(t)x_2(t)x_3(t)$
$= Fx(t)$.

Where $F = (I_3 \otimes M_c)M_r(I_3 \otimes M_i)(I_{27} \otimes M_d)(I_{243} \otimes M_c)W_{[3,27]}M_r(I_9 \otimes M_r)W_{[3,27]}\,M_r(I_9 \otimes W_{[3,3]})$.

By calculating, we can get that

$$F = \delta_{81}\big[\,1 \quad 10 \quad 19 \quad 2 \quad 11 \quad 20 \quad 3 \quad 12 \quad 21 \quad 1 \quad 10 \quad 10 \quad 2 \quad 11 \quad 11 \quad 3 \quad 12 \quad 12$$
$$1 \quad 1 \quad 1 \quad 2 \quad 2 \quad 2 \quad 3 \quad 3 \quad 3 \quad 38 \quad 41 \quad 50 \quad 38 \quad 41 \quad 50 \quad 39 \quad 42 \quad 51 \quad 38 \quad 41 \quad 41$$
$$38 \quad 41 \quad 41 \quad 39 \quad 42 \quad 42 \quad 38 \quad 41 \quad 41 \quad 38 \quad 41 \quad 41 \quad 39 \quad 42 \quad 42 \quad 75 \quad 78 \quad 81 \quad 75$$
$$78 \quad 81 \quad 75 \quad 78 \quad 81 \quad 75 \quad 78 \quad 81 \quad 75 \quad 78 \quad 81 \quad 75 \quad 78 \quad 81 \quad 75 \quad 78 \quad 81 \quad 75$$
$$78 \quad 81 \quad 75 \quad 78 \quad 81 \,\big]$$

and $F^3 = \delta_{81}[\underbrace{1 \cdots 1}_{27} \underbrace{14 \cdots 14}_{27} \underbrace{81 \cdots 81}_{27}]$.

Therefore, $Col(F^3) \subseteq \{\delta_{81}^1, \delta_{81}^{41}, \delta_{81}^{81}\} \subseteq \left\{\delta_{3^4}^1, \delta_{3^4}^{\frac{(2-1)(3^4-1)}{3-1}+1}, \delta_{3^4}^{3^4}\right\} \cdot F$ meets the condition in Theorem 2.1.

The controlled three-valued logic network (10) is inner synchronized for any initial value. And, the iterative process corresponding to different initial values is shown below:

(1) When $x_0 = 0$, the evolution process of the 27 initial states is shown in Table 1:

(2) When $x_0 = 0.5$, the evolution process of the 27 initial states is shown in Table 2:

(3) When $x_0 = 1$, the evolution process of the 27 initial states is shown in Table 3:

In summary, the three-valued logical network (10) is synchronized to 0 when $x_0(t) = 0$; when $x_0(t) = 0.5$ and $x_0(t) = 1$, the three-valued logical network (10) is synchronized to 0.5 and 1 . It can be seen that the three-valued logical network achieves inner synchronization with the same as the driving node. It verifies Theorem 2.1.

**Example 3.2.** Consider a three-valued logical network:

$$\begin{cases} x_1(t+1) = \neg\,(x_1(t) \wedge x_2(t)) \\ x_2(t+1) = x_1(t) \end{cases} \tag{12}$$

Without a driving node, the initial value of this three-valued logical network is now set to (0,0), and the state evolution of each node is as follows:

$(x_1(0), x_2(0)) = \{1, 1\} \rightarrow \{0, 1\} \rightarrow \{1, 0\} \rightarrow \{1, 1\}\cdots$, Obviously the three-valued logical network(12) cannot reach inner synchronization.

By adding a driving node $x_0(t)$, The Eq (12) can be rewritten as

$$\begin{cases} x_0(t+1) = x_0(t) \\ x_1(t+1) = \neg\,(x_0(t) \vee (\neg\,(x_1(t) \wedge x_2(t)))) \\ x_2(t+1) = x_1(t) \end{cases} \tag{13}$$

**Table 1**. Table of node values over time.

| $t_0(x_1, x_2, x_3)$ | $t_1(x_1, x_2, x_3)$ | $t_2(x_1, x_2, x_3)$ | $t_3(x_1, x_2, x_3)$ |
|---|---|---|---|
| 0,0,0 | 0,0,0 | 0,0,0 | 0,0,0 |
| 0,0,0.5 | 0,0.5,0 | 0,0,0 | 0,0,0 |
| 0,0,1 | 0,1,0 | 0,0,0 | 0,0,0 |
| 0,0.5,0 | 0,0,0 | 0,0,0 | 0,0,0 |
| 0,0.5,0.5 | 0,0.5,0 | 0,0,0 | 0,0,0 |
| 0,0.5,1 | 0,1,0 | 0,0,0 | 0,0,0 |
| 0,1,0 | 0,0,0 | 0,0,0 | 0,0,0 |
| 0,1,0.5 | 0,0.5,0 | 0,0,0 | 0,0,0 |
| 0,1,1 | 0,1,0 | 0,0,0 | 0,0,0 |
| 0.5,0,0 | 0,0,0 | 0,0,0 | 0,0,0 |
| 0.5,0,0.5 | 0,0.5,0 | 0,0,0 | 0,0,0 |
| 0.5,0,1 | 0,1,0 | 0,0,0 | 0,0,0 |
| 0.5,0.5,0 | 0,0,0 | 0,0,0 | 0,0,0 |
| 0.5,0.5,0.5 | 0,0.5,0 | 0,0,0 | 0,0,0 |
| 0.5,0.5,1 | 0,1,0 | 0,0,0 | 0,0,0 |
| 0.5,1,0 | 0,0,0 | 0,0,0 | 0,0,0 |
| 0.5,1,0.5 | 0,0.5,0 | 0,0,0 | 0,0,0 |
| 0.5,1,1 | 0,1,0 | 0,0,0 | 0,0,0 |
| 1,0,0 | 0,0,0 | 0,0,0 | 0,0,0 |
| 1,0,0.5 | 0,0.5,0 | 0,0,0 | 0,0,0 |
| 1,0,1 | 0,1,0 | 0,0,0 | 0,0,0 |
| 1,0.5,0 | 0,0,0 | 0,0,0 | 0,0,0 |
| 1,0.5,0.5 | 0,0.5,0 | 0,0,0 | 0,0,0 |
| 1,0.5,1 | 0,1,0 | 0,0,0 | 0,0,0 |
| 1,1,0 | 0,0,0 | 0,0,0 | 0,0,0 |
| 1,1,0.5 | 0,0.5,0 | 0,0,0 | 0,0,0 |
| 1,1,1 | 0,1,0 | 0,0,0 | 0,0,0 |

**Table 2**. Table of node values over time.

| $t_0(x_1,x_2,x_3)$ | $t_1(x_1,x_2,x_3)$ | $t_2(x_1,x_2,x_3)$ | $t_3(x_1,x_2,x_3)$ | $t_4(x_1,x_2,x_3)$ |
|---|---|---|---|---|
| 0,0,0 | 0.5,0.5,0 | 0.5,0.5,0.5 | 0.5,0.5,0.5 | 0.5,0.5,0.5 |
| 0,0,0.5 | 1,0.5,0 | 0,0.5,0.5 | 0.5,0.5,0.5 | 0.5,0.5,0.5 |
| 0,0,1 | 0.5,1,0 | 0.5,0.5,0.5 | 0.5,0.5,0.5 | 0.5,0.5,0.5 |
| 0,0.5,0 | 0.5,0.5,0.5 | 0.5,0.5,0.5 | 0.5,0.5,0.5 | 0.5,0.5,0.5 |
| 0,0.5,0.5 | 0.5,0.5,0.5 | 0.5,0.5,0.5 | 0.5,0.5,0.5 | 0.5,0.5,0.5 |
| 0,0.5,1 | 0.5,1,0.5 | 0.5,0.5,0.5 | 0.5,0.5,0.5 | 0.5,0.5,0.5 |
| 0,1,0 | 0.5,0.5,0.5 | 0.5,0.5,0.5 | 0.5,0.5,0.5 | 0.5,0.5,0.5 |
| 0,1,0.5 | 0,0.5,0 | 0,0,0 | 0.5,0.5,0.5 | 0.5,0.5,0.5 |
| 0,1,1 | 0.5,1,0.5 | 0.5,0.5,0.5 | 0.5,0.5,0.5 | 0.5,0.5,0.5 |
| 0.5,0,0 | 0.5,0.5,0 | 0.5,0.5,0.5 | 0.5,0.5,0.5 | 0.5,0.5,0.5 |
| 0.5,0,0.5 | 0.5,0.5,0 | 0.5,0.5,0.5 | 0.5,0.5,0.5 | 0.5,0.5,0.5 |
| 0.5,0,1 | 0.5,1,0 | 0.5,0.5,0.5 | 0.5,0.5,0.5 | 0.5,0.5,0.5 |
| 0.5,0.5,0 | 0.5,0.5,0.5 | 0.5,0.5,0.5 | 0.5,0.5,0.5 | 0.5,0.5,0.5 |
| 0.5,0.5,0.5 | 0.5,0.5,0.5 | 0.5,0.5,0.5 | 0.5,0.5,0.5 | 0.5,0.5,0.5 |
| 0.5,0.5,1 | 0.5,1,0.5 | 0.5,0.5,0.5 | 0.5,0.5,0.5 | 0.5,0.5,0.5 |
| 0.5,1,0 | 0.5,0.5,0.5 | 0.5,0.5,0.5 | 0.5,0.5,0.5 | 0.5,0.5,0.5 |
| 0.5,1,0.5 | 0.5,0.5,0.5 | 0.5,0.5,0.5 | 0.5,0.5,0.5 | 0.5,0.5,0.5 |
| 0.5,1,1 | 0.5,1,0.5 | 0.5,0.5,0.5 | 0.5,0.5,0.5 | 0.5,0.5,0.5 |
| 1,0,0 | 0,0.5,0 | 0.5,0.5,0.5 | 0.5,0.5,0.5 | 0.5,0.5,0.5 |
| 1,0,0.5 | 0.5,0.5,0 | 0.5,0.5,0.5 | 0.5,0.5,0.5 | 0.5,0.5,0.5 |
| 1,0,1 | 0.5,1,0 | 0.5,0.5,0.5 | 0.5,0.5,0.5 | 0.5,0.5,0.5 |
| 1,0.5,0 | 0.5,0.5,0.5 | 0.5,0.5,0.5 | 0.5,0.5,0.5 | 0.5,0.5,0.5 |
| 1,0.5,0.5 | 0.5,0.5,0.5 | 0.5,0.5,0.5 | 0.5,0.5,0.5 | 0.5,0.5,0.5 |
| 1,0.5,1 | 0.5,1,0.5 | 0.5,0.5,0.5 | 0.5,0.5,0.5 | 0.5,0.5,0.5 |
| 1,1,0 | 0,0.5,0.5 | 0.5,0.5,0.5 | 0.5,0.5,0.5 | 0.5,0.5,0.5 |
| 1,1,0.5 | 0.5,0.5,0.5 | 0.5,0.5,0.5 | 0.5,0.5,0.5 | 0.5,0.5,0.5 |
| 1,1,1 | 0.5,1,0.5 | 0.5,0.5,0.5 | 0.5,0.5,0.5 | 0.5,0.5,0.5 |

Its corresponding algebraic equation is

$$\begin{cases} x_0(t+1) = x_0(t) \\ x_1(t+1) = M_n M_d x_0(t) M_n M_c x_1(t) x_2(t) \\ x_2(t+1) = x_1(t) \end{cases}$$

Let $x(t) = \ltimes_{i=0}^2 x_i(t)$, then $x(t+1) = x_0(t+1)x_1(t+1)x_2(t+1)$

$$= x_0(t)M_n M_d x_0(t) M_n M_c x_1(t) x_2(t) x_1(t)$$
$$= (I_3 \otimes M_n)(I_3 \otimes M_d) M_r (I_3 \otimes M_n)(I_3 \otimes M_c)(I_3 \otimes W_{[3,9]})(I_3 \otimes M_r) x_0(t)x_1(t)x_2(t)$$
$$= Fx(t)$$

The corresponding $F = \delta_{27}[7 \quad 7 \quad 7 \quad 8 \quad 8 \quad 8 \quad 9 \quad 9 \quad 9 \quad 13 \quad 13 \quad 16 \quad 14 \quad 14 \quad 17 \quad 18 \quad 18 \quad 18 \quad 19 \quad 23 \quad 27 \quad 23$ $23 \quad 27 \quad 27 \quad 27 \quad 27]$,

and $F^2 = \delta_{27}[9 \quad 9 \quad 9 \quad 9 \quad 9 \quad 9 \quad 9 \quad 9 \quad 9 \quad 14 \quad 14 \quad 18 \quad 14 \quad 14 \quad 18 \quad 18 \quad 18 \quad 18 \quad 19 \quad 23 \quad 27 \quad 23$ $23 \quad 27 \quad 27 \quad 27 \quad 27]$.

One gets $\mathrm{Col}\,(F^2) \subseteq \{\delta_{27}^9, \delta_{27}^{14}, \delta_{27}^{18}, \delta_{27}^{17}, \delta_{27}^{23}, \delta_{27}^{27}\}$

$$\subseteq \left\{ \delta_{3^2+1}^1, \delta_{3^2+1}^{\frac{3^2+1}{2}}, \delta_{3^2+1}^{3^2}, \delta_{3^2+1}^{3^2+1}, \delta_{3^2+1}^{\frac{3^2+1}{2}+1}, \delta_{3^2+1}^{2\times 3^2}, \delta_{3^2+1}^{2\times 3^2+1}, \delta_{3^2+1}^{\frac{3^2+1+2\times 3^2+1}{2}}, \delta_{3^2+1}^{3^2+1} \right\}.$$

**Table 3**. Table of node values over time.

| $t_0(x_1,x_2,x_3)$ | $t_1(x_1,x_2,x_3)$ | $t_2(x_1,x_2,x_3)$ | $t_3(x_1,x_2,x_3)$ | $t_4(x_1,x_2,x_3)$ |
|---|---|---|---|---|
| 0,0,0 | 1,1,0 | 0,1,1 | 1,1,1 | 1,1,1 |
| 0,0,0.5 | 1,1,0 | 0,1,1 | 1,1,1 | 1,1,1 |
| 0,0,1 | 1,1,0 | 0,1,1 | 1,1,1 | 1,1,1 |
| 0,0.5,0 | 1,1,0.5 | 0.5,1,1 | 1,1,1 | 1,1,1 |
| 0,0.5,0.5 | 1,1,0.5 | 0.5,1,1 | 1,1,1 | 1,1,1 |
| 0,0.5,1 | 1,1,0.5 | 0.5,1,1 | 1,1,1 | 1,1,1 |
| 0,1,0 | 1,1,1 | 1,1,1 | 1,1,1 | 1,1,1 |
| 0,1,0.5 | 1,1,1 | 1,1,1 | 1,1,1 | 1,1,1 |
| 0,1,1 | 1,1,1 | 1,1,1 | 1,1,1 | 1,1,1 |
| 0.5,0,0 | 0.5,1,0 | 0.5,1,1 | 1,1,1 | 1,1,1 |
| 0.5,0,0.5 | 0.5,1,0 | 0.5,1,1 | 1,1,1 | 1,1,1 |
| 0.5,0,1 | 1,1,0 | 0,1,1 | 1,1,1 | 1,1,1 |
| 0.5,0.5,0 | 0.5,1,0.5 | 0.5,1,1 | 1,1,1 | 1,1,1 |
| 0.5,0.5,0.5 | 0.5,1,0.5 | 0.5,1,1 | 1,1,1 | 1,1,1 |
| 0.5,0.5,1 | 1,1,0.5 | 0.5,1,1 | 1,1,1 | 1,1,1 |
| 0.5,1,0 | 0.5,1,1 | 1,1,1 | 1,1,1 | 1,1,1 |
| 0.5,1,0.5 | 0.5,1,1 | 1,1,1 | 1,1,1 | 1,1,1 |
| 0.5,1,1 | 1,1,1 | 1,1,1 | 1,1,1 | 1,1,1 |
| 1,0,0 | 0,1,0 | 1,1,1 | 1,1,1 | 1,1,1 |
| 1,0,0.5 | 0.5,1,0 | 0.5,1,1 | 1,1,1 | 1,1,1 |
| 1,0,1 | 1,1,0 | 1,1,1 | 1,1,1 | 1,1,1 |
| 1,0.5,0 | 0,1,0.5 | 1,1,1 | 1,1,1 | 1,1,1 |
| 1,0.5,0.5 | 0.5,1,0.5 | 0.5,1,1 | 1,1,1 | 1,1,1 |
| 1,0.5,1 | 1,1,0.5 | 1,1,1 | 1,1,1 | 1,1,1 |
| 1,1,0 | 0,1,1 | 1,1,1 | 1,1,1 | 1,1,1 |
| 1,1,0.5 | 0.5,1,1 | 1,1,1 | 1,1,1 | 1,1,1 |
| 1,1,1 | 1,1,1 | 1,1,1 | 1,1,1 | 1,1,1 |

Obviously, the condition in Theorem 2.2 is meet, so the controlled three-valued logical network (13) achieves inner synchronization.

Next, provide the evolution of the network(13) with driving node $x_0(t)$.

(1) When $x_0(t) = 0$, the evolution process of the 9 initial states is shown in Table 4:

(2) When $x_0(t) = 0.5$, the evolution process of the 9 initial states is shown in Table 5:

(3) When $x_0(t) = 1$, the evolution process of the 9 initial states is shown in Table 6:

From the above numerical analysis, it can be seen that the controlled three-valued logical network (13) achieves the inner synchronization, but may not necessarily be synchronized with the driver node. Thus, Theorem 2.2 is verified.

**Table 4**. Table of node values over time.

| $t_0(x_1,x_2)$ | $t_1(x_1,x_2)$ | $t_2(x_1,x_2)$ | $t_3(x_1,x_2)$ |
|---|---|---|---|
| 0,0 | 0,0 | 0,0 | 0,0 |
| 0,0.5 | 0,0 | 0,0 | 0,0 |
| 0,1 | 0,0 | 0,0 | 0,0 |
| 0.5,0 | 0,0.5 | 0,0 | 0,0 |
| 0.5,0.5 | 0.5,0.5 | 0.5,0.5 | 0.5,0.5 |
| 0.5,1 | 0.5,0.5 | 0.5,0.5 | 0.5,0.5 |
| 1,0 | 0,1 | 0,0 | 0,0 |
| 1,0.5 | 0.5,1 | 0.5,0.5 | 0.5,0.5 |
| 1,1 | 1,1 | 1,1 | 1,1 |

**Table 5**. Table of node values over time.

| $t_0(x_1, x_2)$ | $t_1(x_1, x_2)$ | $t_2(x_1, x_2)$ | $t_3(x_1, x_2)$ |
|---|---|---|---|
| 0,0 | 0,0 | 0,0 | 0,0 |
| 0,0.5 | 0,0 | 0,0 | 0,0 |
| 0,1 | 0,0 | 0,0 | 0,0 |
| 0.5,0 | 0,0.5 | 0,0 | 0,0 |
| 0.5,0.5 | 0.5,0.5 | 0.5,0.5 | 0.5,0.5 |
| 0.5,1 | 0.5,0.5 | 0.5,0.5 | 0.5,0.5 |
| 1,0 | 0,1 | 0,0 | 0,0 |
| 1,0.5 | 0.5,1 | 0.5,0.5 | 0.5,0.5 |
| 1,1 | 1,1 | 1,1 | 1,1 |

**Table 6**. Table of node values over time.

| $t_0(x_1, x_2)$ | $t_1(x_1, x_2)$ | $t_2(x_1, x_2)$ | $t_3(x_1, x_2)$ |
|---|---|---|---|
| 0,0 | 0,0 | 0,0 | 0,0 |
| 0,0.5 | 0,0 | 0,0 | 0,0 |
| 0,1 | 0,0 | 0,0 | 0,0 |
| 0.5,0 | 0,0.5 | 0,0 | 0,0 |
| 0.5,0.5 | 0.5,0.5 | 0,0 | 0,0 |
| 0.5,1 | 0,0.5 | 0,0 | 0,0 |
| 1,0 | 0,1 | 0,0 | 0,0 |
| 1,0.5 | 0,1 | 0,0 | 0,0 |
| 1,1 | 0,1 | 0,0 | 0,0 |

## 4 Conclusions

In this paper, two kinds of the inner synchronization of controlled multi-valued logical networks have been investigated using the STP. The first kind achieves synchronization where all nodes synchronize with the driving node. The second kind achieves synchronization among all nodes within the network, but the synchronized state may not be the same as the driving node. Necessary and sufficient conditions for realizing both kinds of the inner synchronization have been derived. Verification has been carried out through two simulation examples, and the results prove the validity of the proposed conditions. In future research, the synchronization problem of controlled multi-valued logical networks with time delays will be further considered.

## Supporting information

**S1 Table. Table of node values over time**. This legend describes the table: the evolution process of node states under various initial states for the simulation examples of Theorem 2.1 and Theorem 2.2.
(PDF)

## Author contributions

**Conceptualization:** Yunyun Deng, Yi Liang.

**Data curation:** Yunyun Deng.

**Formal analysis:** Yunyun Deng, Yi Liang, Xiaolong Qi.

**Funding acquisition:** Yi Liang, Xiaolong Qi.

**Methodology:** Yunyun Deng, Yi Liang.

**Supervision:** Yi Liang.

**Validation:** Yunyun Deng, Yi Liang.

**Visualization:** Xiaolong Qi.

**Writing – original draft:** Yunyun Deng, Yi Liang.

**Writing – review & editing:** Yunyun Deng, Yi Liang, Xiaolong Qi.

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
