## [Decision Letter · Decision Letter 0]

24 Sep 2025

PONE-D-25-32821Inner synchronization of controlled multi-valued logical networksPLOS ONE

Dear Dr. Liang,

Thank you for submitting your manuscript to PLOS ONE. After careful consideration, we feel that it has merit but does not fully meet PLOS ONE’s publication criteria as it currently stands. Therefore, we invite you to submit a revised version of the manuscript that addresses the points raised during the review process.

We look forward to receiving your revised manuscript.

Kind regards,

Yun Shang

Academic Editor

PLOS ONE

Journal Requirements:

“Key Program of Yili Normal University for Comprehensive Strength(No.: 22XKZZ16) Program of the Autonomous Region Tianshan (Youth Top) Talent (No.: 2024TSYCJC0029)

The National Natural Science Foundation of China (No.: 62366054).”

6. We note that your Data Availability Statement is currently as follows: All relevant data are within the manuscript and in Supporting Information files.

7. We note you have included a table to which you do not refer in the text of your manuscript. Please ensure that you refer to Table 2-6 in your text; if accepted, production will need this reference to link the reader to the Table.

Reviewers' comments:

Reviewer's Responses to Questions

**Comments to the Author**

1. Is the manuscript technically sound, and do the data support the conclusions?

Reviewer #1: Yes

Reviewer #2: Partly

2. Has the statistical analysis been performed appropriately and rigorously? 

Reviewer #1: N/A

Reviewer #2: N/A

3. Have the authors made all data underlying the findings in their manuscript fully available?

Reviewer #1: Yes

Reviewer #2: Yes

4. Is the manuscript presented in an intelligible fashion and written in standard English?

Reviewer #1: Yes

Reviewer #2: Yes

5. Review Comments to the Author

Reviewer #1: This paper is devoted to inner synchronization of controlled multi-valued logical networks. Some comments are supplied below to improve the quality.

1. Please enrich Introduction with fault-tolerant optimized control of switched complex networks, synchronization of fractional reaction-diffusion complex networks to highlight the related topics.

2. For (4), how about disturbances imposed on it?

3. Are Definitions borrowed from existing results? If yes, please add theirs sources.

4. The language is poor and there are many syntax errors.

5. Please supply the comparison between the proposed theorems and the existing results on theory and simulation.

Reviewer #2: This paper investigates the inner synchronization of controlled multi-valued logical networks using the semi-tensor product approach. The topic seems to be interesting. However, the current manuscript suffers from significant language issues and a lack of sufficient comparison with existing literature. These issues hinder the clarity and impact of the work. Major revisions are needed before this paper can be considered for publication.

1. The English language usage in the paper needs significant improvement. There are some sentences with unclear or incorrect grammar and syntax, which makes it difficult for readers to understand the intended meaning. For example:

1) In Abstract: "...for the inner synchronization of multi-valued logical networks where the synchronization state is equal to the driving node state..." -> Suggest: "...for the case where the synchronized state of all network nodes equals the state of the driving node..."

2) Page 9: "Research on multi-valued logical networks has been widely applied to networked evolutionary games..." -> Suggest: "Multi-valued logical networks have been widely applied to the study of networked evolutionary games".

3) On the 5th line of Introduction: "...Based on the the above discovery..." -> Suggest: "...Based on the above discovery...".

4) In page 2, "Liu et al. [27] studies..." should be changed to "Liu et al. [27] studied...".

5) Semi-tensor product is abbreviated to STP in page 2, but not it is not used in page 3.

6) In Proposition 1.2, "When" should be changed to "when".

2. The proofs of Theorems 2.1 and 2.2 are hard to follow.

3.　The results of the paper should be compared with other works.

4. Can we studied the fixed-time synchronization of multi-valued logical networks? For fixed-time synchronization you can refer the works: 1) Applied Mathematics and Computation Volume 444, 2023, 127811 2) Journal of Applied Mathematics and Computing (2025) https://doi.org/10.1007/s12190-025-02569-y.

6. PLOS authors have the option to publish the peer review history of their article (what does this mean?). If published, this will include your full peer review and any attached files.

Reviewer #1: No

Reviewer #2: No

---

## [Author Response · Author response to Decision Letter 1]

28 Oct 2025

Dear editor and reviewers,

We read your comments carefully. According to your advice, we made some modifications in the Revised Manuscript with Track Changes and the Manuscript, point by point as follows:

Reviewer #1: This paper is devoted to inner synchronization of controlled multi-valued logical networks. Some comments are supplied below to improve the quality.

TO Reviewer 1:

1. Please enrich Introduction with fault-tolerant optimized control of switched complex networks, synchronization of fractional reaction- diffusion complex networks to highlight the related topics.

Response: In the introduction, we have added content related to switched network optimization control and fractional order complex networks.

2. For (4), how about disturbances imposed on it?

Response: For equation (4), we discussed the issue of network disturbance in two different cases in the Revised Manuscript.

3. Are Definitions borrowed from existing results? If yes, please add theirs sources.

Response: NO. Under the driving and response mechanism, the synchronization definition in this article is the synchronization of a node driving a logical network, while existing synchronization problems related to logical networks are all the synchronization of one logical network driving another logical network.

4.The language is poor and there are many syntax errors.

Response: We have carefully reviewed the entire paper and have made comprehensive revisions to improve the grammar, syntax, and overall clarity. We believe the manuscript is now much clearer and easier to understand.

5. Please supply the comparison between the proposed theorems and the existing results on theory and simulation.

Response:

(1) Theoretical Comparisons: Our work is theoretically distinguished from prior art in the following aspects. While Reference [24] established an inner synchronization theorem for Boolean networks (binary-valued), and the study extended this to multi-valued logical networks, a key limitation remains: many such networks cannot achieve synchronization autonomously. Our primary theoretical contribution is to address this by formulating and solving the problem of achieving inner synchronization via a driving node, a control strategy that forces the network to synchronize.

(2) Simulation Comparisons :The simulation results demonstrate the effectiveness of our method:

The three-valued logical network (3) on Page 4 serves as a case, confirming that the network fails to achieve inner synchronization on its own. This is directly contrasted with the three-valued logical network (10) on Page 9, where the same network (3), when controlled by our proposed driving node, successfully achieves inner synchronization, with all nodes matching the driver's state.

This conclusion is further reinforced by Example 3.2 on Page 10, which shows another network transitioning from a non-synchronized state to a synchronized one upon the introduction of the driving node.

Reviewer #2: This paper investigates the inner synchronization of controlled multi-valued logical networks using the semi-tensor product approach. The topic seems to be interesting. However, the current manuscript suffers from significant language issues and a lack of sufficient comparison with existing literature. These issues hinder the clarity and impact of the work. Major revisions are needed before this paper can be considered for publication.

TO Reviewer 2:

1. The English language usage in the paper needs significant improvement. There are some sentences with unclear or incorrect grammar and syntax, which makes it difficult for readers to understand the intended meaning. For example:

1) In Abstract: "...for the inner synchronization of multi-valued logical networks where the synchronization state is equal to the driving node state..." -> Suggest: "...for the case where the synchronized state of all network nodes equals the state of the driving node..."

2) Page 9: "Research on multi-valued logical networks has been widely applied to networked evolutionary games..." -> Suggest: "Multi-valued logical networks have been widely applied to the study of networked evolutionary games".

3) On the 5th line of Introduction: "...Based on the the above discovery..." -> Suggest: "...Based on the above discovery...".

4) In page 2, "Liu et al. [27] studies..." should be changed to "Liu et al. [27] studied...".

5) Semi-tensor product is abbreviated to STP in page 2, but not it is not used in page 3.

6) In Proposition 1.2, "When" should be changed to "when".

Response: We have carefully reviewed the entire paper and have made comprehensive revisions to improve the grammar, syntax, and overall clarity.

2. The proofs of Theorems 2.1 and 2.2 are hard to follow.

Response: In the process of theorems proof, it mainly involves STP operations, which are computationally complex. We have appropriately simplified the expressions involved in the theorems and added relevant propositions for the proof process.

3. The results of the paper should be compared with other works.

Response:

(1) Theoretical Comparisons: Our work is theoretically distinguished from prior art in the following aspects. While Reference [24] established an inner synchronization theorem for Boolean networks (binary-valued), and the study extended this to multi-valued logical networks, a key limitation remains: many such networks cannot achieve synchronization autonomously. Our primary theoretical contribution is to address this by formulating and solving the problem of achieving inner synchronization via a driving node, a control strategy that forces the network to synchronize.

(2) Simulation Comparisons :The simulation results demonstrate the effectiveness of our method:

The three-valued logical network (3) on Page 4 serves as a case, confirming that the network fails to achieve inner synchronization on its own. This is directly contrasted with the three-valued logical network (10) on Page 9, where the same network (3), when controlled by our proposed driving node, successfully achieves inner synchronization, with all nodes matching the driver's state.

This conclusion is further reinforced by Example 3.2 on Page 10, which shows another network transitioning from a non-synchronized state to a synchronized one upon the introduction of the driving node.

4. Can we studied the fixed-time synchronization of multi-valued logical networks? For fixed-time synchronization you can refer the works: 1) Applied Mathematics and Computation Volume 444, 2023, 127811 2) Journal of Applied Mathematics and Computing (2025) https://doi.org/10.1007/ s12190-025-02569-y.R2

Response It is possible to study the problem of fixed time synchronization in logical networks. Because the state of logical network nodes is finite, synchronization of logical networks is achieved in finite time. For Theorem 2.1 and Theorem 2.2, as long as there exists a specified a (an integer greater than zero) that satisfies Eq.(6) and Eq.(9), the conclusions of fixed time synchronization of logical networks can be obtained separately.

Thanks for your hard work. If the revision is not enough, please contact me, I'd like to follow your advice.

No.: PONE-D-25-32821

Manuscript entitled: Inner synchronization of controlled multi-valued logical networks

Authors: Yunyun Deng, Xiaolong Qi, Yi Liang

---

## [Decision Letter · Decision Letter 1]

29 Dec 2025

Inner synchronization of controlled multi-valued logical networks

PONE-D-25-32821R1

Dear Dr. Yi Liang,

We’re pleased to inform you that your manuscript has been judged scientifically suitable for publication and will be formally accepted for publication once it meets all outstanding technical requirements.

Kind regards,

Yun Shang

Academic Editor

PLOS One

Reviewers' comments:

Reviewer's Responses to Questions

**Comments to the Author**

1. If the authors have adequately addressed your comments raised in a previous round of review and you feel that this manuscript is now acceptable for publication, you may indicate that here to bypass the “Comments to the Author” section, enter your conflict of interest statement in the “Confidential to Editor” section, and submit your "Accept" recommendation.

Reviewer #2: All comments have been addressed

2. Is the manuscript technically sound, and do the data support the conclusions?

Reviewer #2: Yes

3. Has the statistical analysis been performed appropriately and rigorously? 

Reviewer #2: Yes

4. Have the authors made all data underlying the findings in their manuscript fully available?

Reviewer #2: Yes

5. Is the manuscript presented in an intelligible fashion and written in standard English?

Reviewer #2: Yes

6. Review Comments to the Author

Reviewer #2: All of my comments are answered properly. Now I have no further comments and recommend the acceptance of the paper.

7. PLOS authors have the option to publish the peer review history of their article (what does this mean?). If published, this will include your full peer review and any attached files.

Reviewer #2: No

---

## [Editor Report · Acceptance letter]

PONE-D-25-32821R1

PLOS One

Dear Dr. Liang,

I'm pleased to inform you that your manuscript has been deemed suitable for publication in PLOS One. Congratulations! Your manuscript is now being handed over to our production team.

Kind regards,

on behalf of

Dr. Yun Shang

Academic Editor

PLOS One